# Comparison of Transcriptomic Analysis of the Conjunctiva in Glaucoma-Treated Eyes with Dry Eyes and Healthy Controls

**DOI:** 10.3390/biom14010030

**Published:** 2023-12-25

**Authors:** Elena Carnero, Cristina Irigoyen-Bañegil, Itziar Gutiérrez, Leire Extramiana, Alfonso L. Sabater, Javier Moreno-Montañes

**Affiliations:** 1Department of Ophthalmology, Clínica Universidad de Navarra, University of Navarra, 31008 Pamplona, Navarra, Spain; ecarnero@unav.es (E.C.); igutierrezr@unav.es (I.G.); jmoreno@unav.es (J.M.-M.); 2Instituto de Investigación Sanitaria de Navarra (IdiSNA), 31008 Pamplona, Navarra, Spain; lextramiana@unav.es; 3Department of Ophthalmology, Ocular Surface Center, Bascom Palmer Eye Institute, Miami, FL 33136, USA; asabater@med.miami.edu

**Keywords:** impression cytology, glaucoma medication, ocular surface disease, RNA sequencing

## Abstract

Ocular surface disease (OSD) associated with topical glaucoma drugs is a common issue impacting treatment adherence. We aimed to identify conjunctival transcriptomic changes in glaucoma and dry eye patients, comparing them to healthy controls. Bulbar conjunctival specimens were collected via impression cytology from 33 patients treated for glaucoma, 9 patients with dry eye, and 14 healthy controls. RNA extraction and bulk RNA sequencing were performed, followed by bioinformatics analysis to detect gene dysregulation. Ingenuity pathways analysis (IPA) identified pathways and biological processes associated with these transcriptomic changes. Sequencing analysis revealed 200 modified genes in glaucoma patients compared to healthy individuals, 233 differentially expressed genes in dry eye patients versus controls, and 650 genes in treated versus dry eye samples. In glaucoma patients, 79% of altered pathways were related to host defense, while dry eye patients showed a 39% involvement of host response, 15% in cellular proliferation and integrity, and 16% of mitochondrial dysfunction. These findings were validated through qRT-PCR. Glaucoma patients showed an intensified conjunctival immune response as a potential cause of OSD, whereas in dry eye patients, in addition to the immune response, other mechanisms such as mitochondrial dysfunction or reduced cellular proliferation were observed.

## 1. Introduction

Ocular surface homeostasis and integrity are the major contributors to ocular protection. Numerous factors affect the ocular surface, i.e., “internal” factors, such as age and autoimmune or metabolic diseases, and “external” factors, such as infections, contamination, or contact lens wear. Moreover, the ocular surface is the gateway for antibiotics, steroids, and glaucoma ocular drops. The conjunctiva comprises an unkeratinized epithelial layer in which other cellular types reside, such as goblet cells (GCs), which prevent pathogenic adhesion and invasion, trap debris, and provide lubrication. Mucin secretion by GCs is regulated by the nervous system, and is altered by ocular surface conditions and inflammatory cytokines [1]. Immune cells infiltrate the conjunctiva, i.e., primarily dendritic cells (DCs), T cells, and neutrophils. The internal and external factors can induce damage and stress in the conjunctival epithelium, triggering production of inflammatory mediators, which initiate the inflammatory cascade by immune cell recruitment and maturation. These mediators, combined with exposure to autoantigens, can lead to an adaptive T-cell-mediated response. Recent studies have reported the potential role of GCs in the immune tolerance of the ocular surface by modulating antigen delivery and the antigen-specific immune response [1,2,3].

Glaucoma treatment includes hypotensive drugs to control the intraocular pressure (IOP), the only treatable risk factor to prevent visual loss [4,5]. Several types of IOP-lowering eye drops are available commercially: prostaglandin analogues, beta-blockers, carbonic anhydrase inhibitors (CAIs), and sympathomimetic agents. Prostaglandin analogues increase aqueous humor drainage, and the other IOP-lowering treatments inhibit production. A fifth group, Rho-kinase inhibitors, recently approved for use in patients, are believed to improve aqueous drainage through the trabecular meshwork by acting on the cytoskeleton [6]. These treatments require daily application of a varying number of drops, depending on the type and/or number of medications; 40% of patients with glaucoma are estimated to need more than one type of IOP-lowering medication [7]. Patients treated for glaucoma frequently present with symptoms associated with ocular surface disease (OSD) [8,9], the prevalence of which is estimated to be about 15% among people older than 65 years and 59% in patients with glaucoma [10]. OSD associated with glaucoma treatment causes decreased quality of life and is associated with non-compliance with treatment, leading to disease progression [11]. OSD in treated patients has been associated with preservatives in the drugs, specifically benzalkonium chloride (BAK) [12]. BAK is associated with adverse effects such as ocular irritation, tear film instability, chronic inflammation, and subconjunctival fibrosis. Mohammed et al. evaluated the effects of different preservatives in treatment-naïve patients, profiling temporal changes in cytokines in impression cytology and tear samples. They saw that prolonged use of BAK-preserved drops correlated with elevated cytokine levels and ocular discomfort, while patients using polyquad (PQ)-preserved drops also showed moderately high cytokine levels, suggesting potential subclinical inflammation [13]. Although it is undeniable that BAK-free formulations cause less toxicity, toxicities associated with the active ingredients or other excipients also must be considered. In addition, the preservative toxicity can differ and be greater when used alone compared with being an ingredient in a formulation [14,15]. Thus, β-blockers have been associated with burning, prickling, foreign-body sensation, and ocular redness [16]. These compounds may increase production of reactive oxygen species (ROS) in the conjunctival epithelium [17] and exert a cytotoxic effect on the corneal epithelium [18]. Sympathomimetics, especially brimonidine, are associated with allergic reactions and conjunctival edema [19,20]. The toxicity associated with CAIs is predominantly associated with corneal endothelial cell dysfunction [21]. Finally, prostaglandins induce conjunctival hyperemia not associated with inflammation [22,23]. Patients with glaucoma have increased expression of inflammatory markers such as human leukocyte antigen-DR and interleukin (IL)-6 in combinations of prostaglandins and β-topical blockers at the ocular surface [24]. Different cytokines, including several interleukins, macrophage inflammatory protein 1 alpha, or fibroblast growth factor-β, have been detected in tears of patients treated for glaucoma, indicating that inflammation is a key factor affecting the ocular surface in these patients [25,26,27]. Inflammation always results from an insult to the ocular surface. 

According to the 2017 Dry Eye Workshop, dry eye is a multifactorial disease of the ocular surface characterized by a loss of homeostasis of the tear film, accompanied by ocular symptoms. In this disease, instability, hyperosmolarity of the tear film, inflammation, damage to the ocular surface, and neurosensory abnormalities play important etiological roles [28]. This syndrome is divided into two main groups: evaporative dry eye and hyposecretory dry eye. In the evaporative subgroup, the primary alteration is found in the lipid layer of the tear film, while in the hyposecretory subgroup, the main alterations occur in the mucin and aqueous layers of the tear film [28,29].

In a recent study by Mauduit et al., the researchers examined dysregulated pathways in nonobese diabetic (NOD) mice, a well-established model for investigating the development of Sjögren’s syndrome (SS), one of the leading causes of aqueous-deficiency dry eye. The most significant pathway up-regulated was the “TYROBP Causal Network”, which had not been described previously in SS. This pathway is linked to macrophage activity, chemokine signaling, microglia phagocytosis, and apoptosis. Additionally, they observed a significant decrease in genes related to amino acid and lipid metabolism, along with ATP production, suggesting a mechanism contributing to the progression of the disease in these mice [30]. Changes in the composition of meibum in individuals with Meibomian gland dysfunction (MGD) are thought to alter microbial populations, contributing to the activation and increased abundance of immune cells, thereby leading to an inflammatory response in tears and glandular tissue. This cascade of events impacts the differentiation of glandular cells and their capacity to produce and release lipids, which is considered critical in initiating pathological changes [29,31].

Furthermore, Suárez-Cortés et al. recently compared implicated pathways in dry eye disease (DED) and ocular allergy (OA), discovering shared features like chronic inflammation and alterations in the tear film mucous layer. Both groups exhibited common biomarkers (e.g., Th17, Th1, and Th2 cells) and overexpression of proinflammatory cytokines (e.g., IL-1α, IL-1β, IL-17, TNFα, INFγ, IL-4, IL-5, and IL-13) in tears and on the ocular surface. Additionally, they shared inflammatory responses involving metalloproteinases (MMPs) and tissue inhibitors (TIMPs). Despite similarities in ocular surface inflammation, triggers of the disease differed: low tear production in DED led to instability and cascading inflammatory events, while OA involved eosinophils, mast cells, neutrophils, and Th2 lymphocytes [32].

We hypothesize that ocular surface inflammation secondary to toxicity from BAK results from molecular and cellular pathways that are different from other ocular surface inflammatory conditions such as dry eye disease, despite both groups presenting similar symptoms. To test this hypothesis, we performed transcriptomic analysis of samples from the conjunctiva of patients on glaucoma medications containing BAK, and compared these to samples from dry eye patients and healthy controls. The ultimate goal is to identify new targets for the treatment of these patients, providing a more personalized approach to each disease.

## 2. Materials and Methods

### 2.1. Study Design

A cross-sectional study was conducted involving patients diagnosed with dry eye syndrome, patients using glaucoma hypotensive drops, and a control group.

For the dry eye group, the inclusion criterion consisted of individuals with a clinical diagnosis of dry eye according to the DEWS classification [28]. The exclusion criteria for this group included patients receiving topically administered medications containing preservatives (e.g., for glaucoma, blepharitis, etc.); furthermore, individuals with a history of previous ocular surgeries affecting the ocular surface (refractive surgery, eyelid surgery, corneal surgery), use of systemic medication that may cause alterations in tear production, use of contact lenses, or eyelid pathology (lagophthalmos, exophthalmos, ectropion, etc.) were also excluded from the dry eye group.

For glaucoma-treated patients, the inclusion criteria involved individuals on hypotensive drops for at least 1 year, including prostaglandin analogues, beta-blockers, carbonic anhydrase inhibitors, and alpha agonists alone or in combinations (Appendix A). Exclusion criteria for this group comprised individuals receiving systemic or topical medication that could alter results (e.g., anti-inflammatory or immunosuppressant drugs), as well as those with a history of previous eye surgery that could alter the ocular surface.

The control group comprised asymptomatic individuals consecutively recruited from hospital staff, nurses, patient relatives, and those referred for routine visual acuity examinations without ocular diseases. The inclusion criteria for the control group were an intraocular pressure (IOP) of 20 mmHg or lower, normal visual fields, and the absence of a familial history of glaucoma. All participants were of European Caucasian descent, had no corneal or retinal pathology, no significant media opacity obscuring the eye fundus, no history of amblyopia, and no contraindication to dilation. Additionally, they showed no intolerance to topical anesthetics, mydriatic agents, or fluorescein dye.

### 2.2. Conjunctival Impression Cytology

Conjunctival impression cytology (CIC) was performed using an Eyeprim device (OPIA Technologies, Paris, France), following the manufacturer’s instructions. Topical anesthesia was instilled into the lower conjunctival sac. The device was placed onto the temporal bulbar conjunctiva. A total of 140 samples from 70 patients were collected, with one sample obtained from each eye. Out of the total 140 collected conjunctival impression cytology (CIC) samples, 56 were selected for RNA sequencing; 33 patients were treated for glaucoma, 9 patients had dry eye, and 14 patients were healthy (Table 1). Only one eye from each patient was used for RNA sequencing. The remaining 95 samples underwent quality classification, with 30 of them used for result validation (Appendix A). The dry eye group included patients who had been diagnosed with dry eye unrelated to glaucoma treatment and who used artificial tears. The treated group included patients diagnosed with bilateral open-angle glaucoma who had been treated with a minimum of one medication for at least 1 year. The numbers and types of treatments for each patient were recorded (see Appendix A). The control group included subjects recruited among hospital staff and patients referred for a routine visual acuity examination with no ocular diseases or familial glaucoma history or use of topical drops for glaucoma or dry eye for at least 1 year. All of the patients provided written informed consent before the study began.

### 2.3. RNA Isolation, Quantification, and Reverse-Transcriptase Polymerase Chain Reaction

The RNA was isolated from the CIC using the Maxwell^®^ 16 RNA extraction kit (Promega, Madison, WI, USA), following the manufacturer’s instructions. The RNA quantification was evaluated via the NanoDrop^®^ spectrophotometer (Thermo Fisher Scientific, Waltham, MA, USA). The quality of the RNA was evaluated using the Bioanalyzer (Agilent Technologies, Lexington, MA, USA). 

Quantitative RT-PCR (qRT-PCR) was conducted using the C1000 Touch Thermal Cycler from Bio-Rad. The samples were initially incubated at 37 °C for 60 min, then at 95 °C for 60 s, and then immediately cooled to 4 °C. Subsequent qPCR steps were carried out in the CFX96 Real-Time System from Bio-Rad, following the protocol detailed by Carnero et al. [33]. The resulting data were analyzed with Bio-Rad CFX Manager software 3.1. GAPDH levels were assessed as a reference in all cases, and only samples with comparable GAPDH amplification underwent further analysis. The relative RNA levels were determined using the 2−ΔCt method, where ΔCt represents the difference between the Ct values of the gene of interest and the GAPDH mRNA internal control.

### 2.4. RNA Sequencing and Data Analysis

Bulk RNA sequencing was performed following the metal artifact reduction sequence protocol [34] adapted for bulk RNA sequencing [35] with minor modifications using the NextSeq 500 (Illumina Inc., San Diego, CA, USA) with dual-index sequencing (Rd1, 68 cycles; Rd2, 15 cycles; i7, 8 cycles) at a depth of 10 million reads/sample. 

RNA sequencing data analysis was performed using the following workflow: (1) the quality of the samples was verified using FastQC software 0.12.0 (Babraham Institute; Cambridge, UK); (2) the alignment of reads to the human genome (hg38) was performed using STAR software 2.7.0a [36]; (3) gene expression quantification using read counts of exonic gene regions was performed with featureCounts 2.0.6 [37]; (4) the gene annotation reference was Gencode v29 [38]; and (5) differential expression statistical analysis was performed using R/Bioconductor 3.18 [39]. First, the gene expression data were normalized with edgeR 3.18 [40] and voom [41]. After quality assessment and outlier detection using R/Bioconductor 3.18 [39], filtering was performed. Linear models for microarray data [41] identified the genes with significant differential expressions between groups. Genes were selected as differentially expressed using a *p*-value cut-off of <0.01. Biologic knowledge extraction was complemented using Ingenuity Pathway Analysis (IPA) (Ingenuity Systems, Redwood City, CA, USA), the database of which includes manually curated and traceable data derived from literature sources. 

### 2.5. Data Analysis

Both *p*-values and Z-scores are statistical measures used in bioinformatics and other fields, often in the context of analyzing high-throughput data such as gene expression data.

A *p*-value is a measure of the evidence against a null hypothesis. In the context of gene expression analysis, it helps researchers assess whether the observed differences in expression between experimental groups are statistically significant. A low *p*-value (typically below a chosen significance threshold, e.g., 0.05) suggests that there is enough evidence to reject the null hypothesis, indicating a potentially meaningful difference. In the context of differential gene expression analysis, a *p*-value is often associated with statistical tests such as the *t*-test or ANOVA.

The Z-score is a standardized score that represents the number of standard deviations a data point is from the mean of a population. In the context of bioinformatics and pathway analysis tools like Ingenuity Pathway Analysis (IPA), Z-scores can be used to assess the activation or inhibition status of biological pathways. For example, in IPA, a positive Z-score may indicate pathway activation, while a negative Z-score may suggest pathway inhibition. Z-scores help interpret whether the observed gene expression changes are consistent with the expected direction of change based on prior knowledge of pathway regulation.

The Shapiro–Wilk test was employed to assess whether the samples deviated significantly from a normal distribution. Differences between the severe and non-severe groups were examined using Student’s *t*-test for parametric data, while the Mann–Whitney U-test was utilized for non-parametric parameters. To identify associations among parametric variables, Pearson’s correlation coefficient was calculated for normal distributions, whereas Spearman’s correlation coefficient was applied for non-parametric variables. The statistical analysis was conducted using SPSS software version 20.0.1 (SPSS Inc., Chicago, IL, USA).

## 3. Results

### 3.1. Differential Gene Expression Analysis Using RNA Sequencing Data Derived from Treated and Dry Eye Patients Compared to Healthy Individuals

Sequencing analysis detected the expression of 58,722 genes annotated in reliable databases, 200 of which were modified in patients treated for glaucoma compared to healthy individuals (logFC > 0.5; *p* < 0.01) (Table 2). Following the same selection criteria, we detected the differential expression of 233 genes compared with dry eye and control samples, and 650 genes were differentially expressed in treated glaucoma patients compared to dry eye samples (logFC > 0.5; *p* < 0.01) (Table 2). Volcano plots that were used to assess the messenger RNA expression variation between groups initially showed significant differentially expressed messenger RNAs (logFC > 1, *p* < 0.01) (Figure 1).

### 3.2. Genes and Pathways Dysregulated in Eyes Treated with Topical Glaucoma Medications

Overall, the gene expression profiles in patients treated for glaucoma were compared to control patients using Qiagen Ingenuity Pathway Analysis (IPA). IPA is bioinformatics software (2018 version) that helps researchers analyze and interpret omics data, such as gene expression, microarray, and RNA-seq data. It is commonly used for pathway analysis, network analysis, and functional annotation to gain insights into the biological significance of experimental results.

When IPA analysis was performed, a marked activation of pathways implicated in the immune response was seen. Host defense-related pathways represented 79% of the total pathways (Table 3). Increases in the expressions of different cellular damage sensors or their components were identified: *Toll-like receptor 4 (TLR4*) (*p* < 0.001), *LY86* (*p* < 0.001), or *P2RX7* (*p* = 0.002) in patients treated for glaucoma; elevated expression of genes related to antigen-presenting cells were identified in dendritic cell activation: *TREM2* (*p* = 0.003), *CD86* (*p* < 0.001), *FCER1G* (*p* = 0.009), and *FCGR3A* or *TYROBP* (*p* < 0.001). Induced expressions of Tec and Src kinase members that act immediately downstream of antigen and Fc receptors in immune cells were also found: *BTK* (*p* = 0.002), *HCK* (*p* = 0.004), and *FGR* (*p* = 0.005) (Figure 2). Increases in these molecules were also seen to a lesser extent in dry eye genes versus controls. The other pathways altered in patients with glaucoma were related to control cellular proliferation and differentiation (Table 3).

### 3.3. Routes and Genes Dysregulated in Dry Eye

In dry eye, 39% of altered pathways were related to cellular growth and differentiation; that is, there were decreases in proteins that participate in the hippo pathway, a core pathway regulating organ size, cellular proliferation, and differentiation that includes transcription factors *YAP1* (*p* = 0.009), *TEAD1* (*p* = 0.005), and *SMAD4* (*p* = 0.006). Several components of the transforming growth factor β (TGF-β) pathway, which acts on both autocrine and paracrine activities to regulate cellular growth, differentiation, and migration, were also down-regulated in patients with dry eye. Decreased expression of *ITGB8* that regulates *TGF-β* activation, besides other integrin-related genes, was also observed. However, increased activity of pathways associated with mitochondrial dysfunction and the *Nrf2*-mediated oxidative stress response (Figure 3) was seen. Mitochondrial genes such as *MT-CYB* or *MT-CO2* were overexpressed, as were genes associated with oxidoreductase activity; *SOD* genes and *GPX2* were up-regulated in patients with dry eye. The expression of *SIRT*, a deacetylase involved in antioxidant activity, DNA repair, and aging, was regulated by *ROS* that were down-regulated in dry eye (Figure 3). Finally, genes such as *RGS1*, *RNASE6*, or *FCER1G* that participate in the host defense mechanism, comprised 39% of the total pathways (Table 3).

### 3.4. Differential Pathways Dysregulated in Patients Treated Topically for Glaucoma Compared with Patients with Dry Eye

We evaluated the differences in the cellular responses of patients who were treated and those with dry eye. Major differences were found in the regulation of cellular adhesion, migration, and maintenance of epithelial integrity. These routes were under-expressed in patients with dry eye compared to those treated for glaucoma (Table 4). We observed higher expressions of the routes related to oxidative stress responses in patients with dry eye compared to patients treated for glaucoma. However, we found that treated patients had increases in pathways related to the immune system compared to patients with dry eye and with controls (Table 4).

## 4. Discussion

The current study investigated the cellular mechanism underlying the clinical symptoms of patients treated for glaucoma and patients with dry eye syndrome, and whether the cellular responses in both groups of patients were similar. We performed CIC on the bulbar conjunctival samples obtained from patients treated topically for glaucoma, from patients diagnosed with dry eye syndrome, and from healthy controls. 

### 4.1. Differences between Dry Eye and Controls

The main alterations in the dry eye syndrome group were observed in conjunctival epithelial cells (proliferation, cellular adhesion, migration, differentiation), ocular surface immunotolerance, oxidative stress, inflammation, and mitochondrial DNA damage.

When comparing the ocular surface of patients with dry eye syndrome to that of individuals with healthy conjunctiva, we observed a significant dysregulation of gene expression and pathways associated with proliferation and tissue differentiation. Several genes of the hippo pathway, including *YAP1*, *TEAD1*, or *SMAD4*, were down-regulated in patients with dry eye. This route regulates human conjunctival epithelial cellular proliferation and cell attachment, and is closely related to the *TGF*-*β* signaling pathway [42], which plays a key role in cellular adhesion, migration, proliferation, and differentiation [43]. 

Moreover, *TGF-β* secreted by human lacrimal glands, corneal and conjunctiva epithelia [44,45], and presumably immune system cells [46] has an important role in preventing unwanted immune responses by controlling the tolerance against self and innocuous antigens through generation of regulatory T cells (Tregs) and impairing the clonal expansion and functions of T cells, B cells, and natural killer cells or DCs [47]. 

Recently, Stockis and colleagues reported that integrin *αVβ8* is indispensable for *TGF-β1* activation from *GARP*/latent *TGF-β1* complexes on the surface of human Tregs. *ITGB1* and especially *ITGB8* are down-regulated in patients with dry eye [48]. These data suggest that ocular surface immunotolerance decreases in these patients. Thus, the ocular surface would have increased inflammation, as we and other authors observed [49]. 

In the results of the dry eye syndrome group, we also observed an increase in *SOD1*, responsible for 90% of *SOD* antioxidant activity. Mice lacking this gene showed increased lipid and DNA damage related to oxidative stress and increased immune cells infiltration, decreased GC density, and lacrimal gland dysfunction [50]. Together with *SOD1*, we found elevated expressions of other genes associated with an antioxidant response and related to *NRF2*, a transcription factor and master regulator of antioxidant gene expression. 

With aging, mitochondrial DNA (mtDNA) mutation accumulates, and the mitochondrial structure is disrupted. The response to mtDNA damage was reported to increase the mtRNA copy number [51], and we observed an increase in mtRNA in patients with dry eye compared to healthy eyes. 

Patients with dry eye in this study also had a significant decrease in *SIRT1* RNA levels compared to controls. *SIRT1* is involved in the response to chronic inflammation; it regulates the redox environment and compensates for oxidative damage by converting NAD to NADH; it is highly conserved through evolution; and it regulates multiple cellular metabolism components regarding aging, DNA repair, mitochondrial biogenesis, or apoptosis [52,53,54,55]. *SIRT1* has an important role in ocular morphogenesis and retinal development, and is associated with cataracts, corneal diseases, age-related macular degeneration, diabetic retinopathy, glaucoma, and optic neuritis [56,57,58]. 

Findings in the dry eye syndrome group revealed distinctive patterns when compared with the control groups. We observed that the conjunctiva of patients with dry eye has a decrease in the genes implicated in the maintenance of immunotolerance, an increase in redox balance pathways that indicate increased levels of *ROS* and is implicated in increased inflammation (Figure 4). Inflammation is a major contributor to both the appearance of dry eye disease and its progression over time, although the immunologic mechanisms resulting in the pathogenesis have yet to be elucidated. Stressors, including environmental hazards such as toxins or infections, endogenous stress related to age, and genetic factors, can disrupt the fine-tuned balances on the ocular surface and trigger the inflammatory response. 

### 4.2. Differences between Glaucoma-Treated Patients and Controls

When it comes to treating glaucoma patients, in addition to the factors described previously, the daily use of topical treatments must be considered. We already evaluated the clinical signs of damage on the ocular surface of these patients [8]. Since most glaucoma treatments contain BAK, and numerous studies have reported its role as a pro-inflammatory, pro-apoptotic, and pro-oxidative agent, BAK is assumed to be the primary damaging agent for the ocular surface. This study emphasizes the activation of immune response-related pathways, such as dendritic cells and inflammasome-related pathways, and the potential role of specific receptors and kinases in the observed changes to ocular surface damage mediated by glaucoma drugs.

Among the routes dysregulated in the treated glaucoma patients, we observed activation of the *TREM* pathway. *TREM* receptors are expressed on immune cells, such as neutrophils, mature monocytes, macrophages, DCs, or platelets. Specifically, *TREM2* coupled with *DAP-12/TYROBP* was highly up-regulated in our results, and is involved in survival, migration, and maturation of DCs to secondary lymphoid organs [59]. 

We also detected a high expression of *CD86* in these patients, a dendritic marker with low expression under normal conditions; however, it is up-regulated after DC activation and has an important role in priming naïve T cells [59]. Finally, *Cd11C* (*ITGAX*), a membrane marker found at high levels in DCs among other cells, was overexpressed in patients treated for glaucoma. 

We also found clear up-regulation of *TLR4* in glaucoma patients, which seemed to be a major regulator of chronic inflammation and inflammation-related diseases; its downstream signaling with the *nuclear factor-κβ* (*NF-κβ*) is critical for the expression of inflammatory cytokines. Zettel et al. reported that *TLR4* expression in DC is required for increases in inflammatory cytokines, and promotes inflammation after liver damage [60]. We speculated that treatment-related damage to the ocular surface may increase the presence of DCs, and *TLR4* could participate in the activation of DC. These results suggest an increase in activated DCs in the bulbar conjunctiva of treated patients. 

The *TLR4* pathway participates in the activation of inflammasomes, which are protein complexes of the innate immune system involved in activating the inflammatory response [61,62]. Briefly, inflammasomes require two steps of activation; the first is a priming phase in which a sensor such as *TLR4* recognizes damage or stress signals [63]. Several receptors have been described to participate in the inflammasome activation phase, the best known of which is *P2RX7* (*P2X7*), which is significantly up-regulated in patients treated for glaucoma (Figure 5). The *P2X7* receptor is an ion channel activated by ATP. BAK and preserved prostaglandin analogues activate the *P2X7* receptor due to ATP release, leading to the death of corneal and conjunctival cells [64,65,66]. In addition, an increase in NAD+ lowers the ATP threshold for *P2XR7* activation; we detected decreased *CD38* (logFC = −0.8; *p* = 0.004), an enzyme that degrades NAD+ [67]. ATP levels are increased in human corneal and conjunctival epithelial cells exposed to hyperosmotic challenge and in patients with dry eye [68]. However, those authors did not relate the damage to activation of *P2X7*; they suggested another receptor as mediator of the damage in this case. Interestingly, we did not find any difference in *P2X7* levels between patients with dry eye and controls, but it is highly expressed specifically in patients treated for glaucoma.

Recently, *BTK*, a member of the TEC kinases that is overexpressed in treated patients (Figure 5), was described as an essential player in inflammasome activation in macrophages. *BTK* participates in the bonding of the inflammasome components necessary for the activation of caspase 1 and subsequent secretion of the active forms of IL1 and IL18 [69]. Thus, we observed high up-regulation of three key inflammasome activators, *TLR4*, *P2X7*, and *BTK*. Further studies should determine the role of this complex in the pathogeny associated with use of glaucoma eye drops and their components.

### 4.3. Study Limitations

RNA sequencing of impression cytology (CIC) provided an overview of the overall gene expression in the entire cell set at the bulbar conjunctival level in the two study groups that exhibited similar clinical symptoms. However, a study limitation was that we could not distinguish the expressions of the different cellular types, except for specific markers. Another limitation was the small number of patients; nonetheless, we studied several key genes using various CIC samples collected for the study, comprising 9 controls, 8 dry eye, and 30 treated samples (Appendix A). We validated this information using 30 samples out of the 140 we collected that were not previously used, in order to increase technical reproducibility (Appendix A).

## 5. Conclusions

After a literature review, we acknowledged research about inflammatory markers in impression cytology samples using NanoString^®^ nCounter technology, which is based on an inflammatory human code set containing 249 inflammatory genes [70,71]. We also acknowledged studies in mice and in vitro studies involving corneal cells [72]. Additionally, we identified studies on microbiome aspects [73], although they were somewhat distant from the focus of our current investigation. It is worth noting that there are proteomic studies available on tears from patients undergoing antiglaucomatous treatment [72]. We also found studies on the damage to the conjunctiva and Tenon’s layer in rabbits, as well as studies using cell lines or bulk gene expression data related to steroid response [74]. Nevertheless, to the best of our knowledge, this study represents the first analysis of cytological samples from the conjunctiva of patients, and the first to conduct bulk RNA sequencing in individuals with glaucoma and dry eye, comparing the findings with those from healthy subjects.

This additional dimension in the literature contributes to a more comprehensive understanding of the ocular environment in individuals receiving such treatment, complementing our exploration of gene expression in conjunctival cytological samples.

In conclusion, ocular surface disease in patients with dry eye syndrome and those related to glaucoma treatment exhibit shared clinical symptoms, as previously reported by our group and others. Although these symptoms have been linked to an inflammatory response, the exact mechanism triggering this inflammation has not been fully elucidated. Transcriptome analysis in these patients revealed the involvement of several signaling pathways in the initiation of the inflammatory response, and these pathways may differ between dry eye and glaucoma. A deeper understanding of the molecular mechanisms causing damage could unveil insights into the molecules responsible for mediating the damage, paving the way for new approaches to enhance the treatment of ocular surface pathologies in both dry eye and glaucoma. 

## Figures and Tables

**Figure 1 biomolecules-14-00030-f001:**
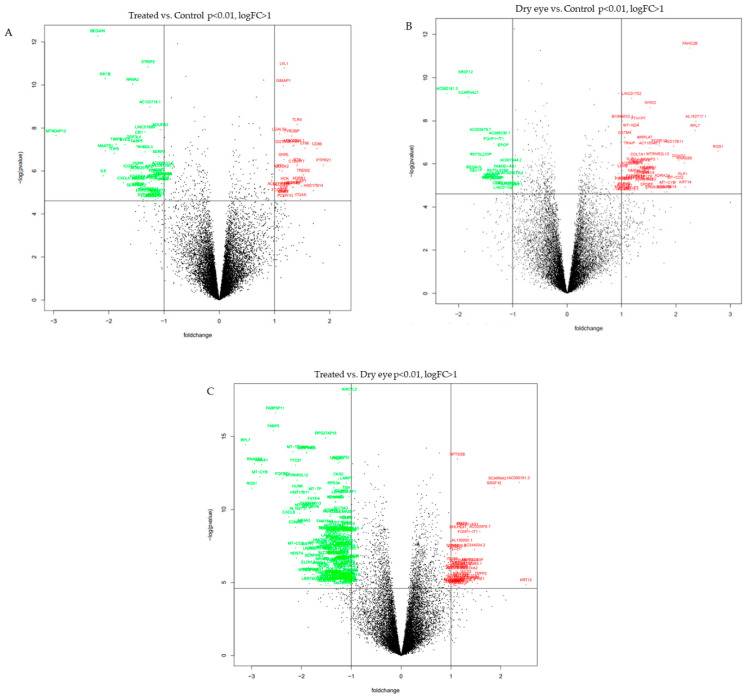
Volcano plots display differentially expressed genes between (**A**), treated versus control patients, (**B**), dry eye versus control patients, and (**C**), treated versus patients with dry eyes. The y-axis corresponds to the mean expression value of -log (*p*-value), and the x-axis displays the fold change value. The red genes represent the up-regulated expressed transcripts; the green genes represent the down-regulated transcripts.

**Figure 2 biomolecules-14-00030-f002:**
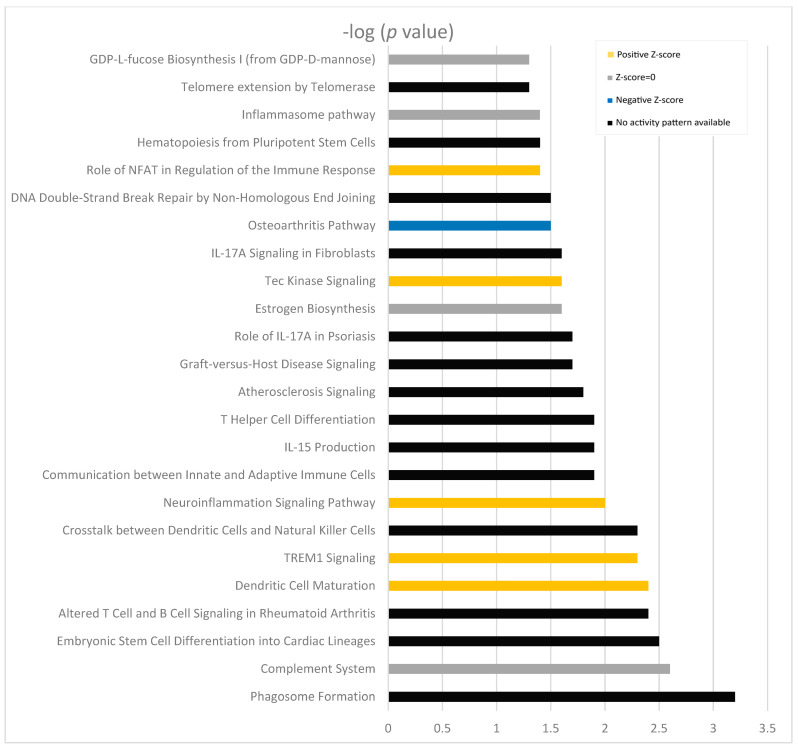
Pathways altered in the functional analysis of dysregulated genes in treated patients compared to healthy individuals using IPA. Orange bars indicate activation of the route (positive Z-score), blue bars predicted a decreasing activity of the pathway (negative Z-score). Grey lines corresponding to the grey pathways represent an effect not predicted based on the IPA activation Z-scores.

**Figure 3 biomolecules-14-00030-f003:**
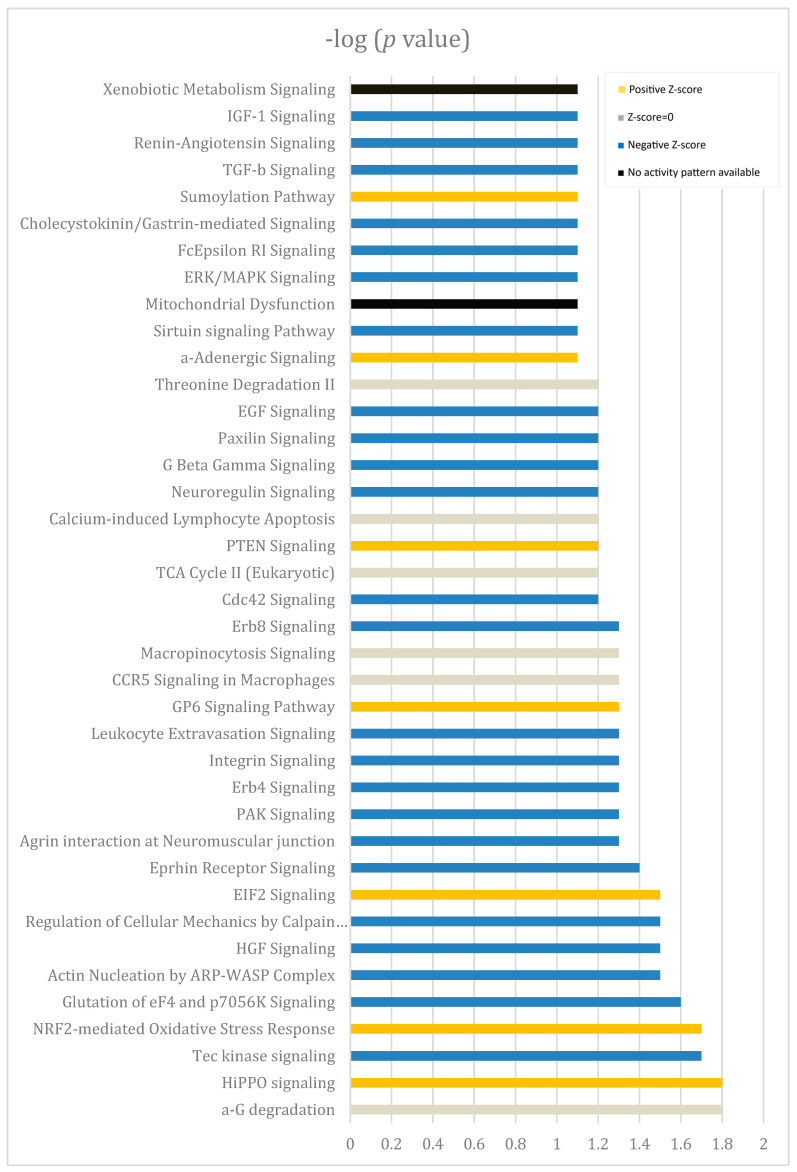
Pathways altered in the functional analysis of dysregulated genes in treated patients compared to healthy individuals using IPA. Orange bars indicate activation of the route (positive Z-score), and blue bars predict decreasing activity of the pathway (negative Z-score). Grey lines correspond to grey pathways representing an effect not predicted based on the IPA activation Z-scores.

**Figure 4 biomolecules-14-00030-f004:**
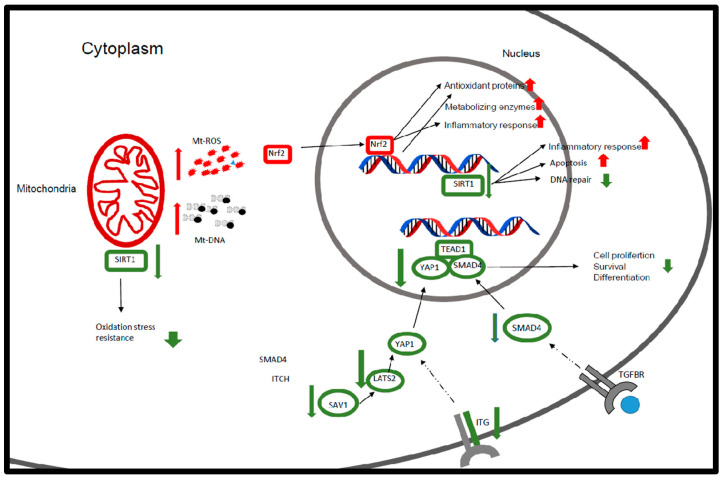
Schematic representation of genes and pathways affected in patients with dry eye based on IPA pathways. Genes shown in the figure were obtained from the RNA-seq data in this study.

**Figure 5 biomolecules-14-00030-f005:**
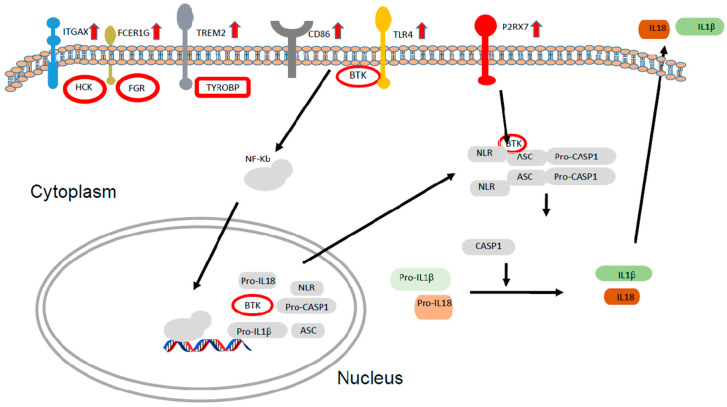
Schematic representation of genes and pathways affected in patients treated for glaucoma based on IPA pathways. Genes shown in the figure were obtained from the RNA-seq data in this study.

**Table 1 biomolecules-14-00030-t001:** Demographic data of patients included in the sequencing study.

	Total	Sex M/F ^1^	Age (Range) ± SD
Control	14	6/8	58.32 (32–79) ± 14.58
Treated	33	19/14	66.32 (29–89) ± 15.30
Dry eye	9	3/6	70.04 (61–77) ± 5.33
Total	56	28/28	66.75 (29–89) ± 14.45

^1^ M/F: Male/Female.

**Table 2 biomolecules-14-00030-t002:** Numbers and types of dysregulated RNA.

	Dysregulated Genes	Up	Down
Treated vs. control	200	86	114
Dry eye vs. control	233	107	126
Treated vs. dry eye	650	259	391

**Table 3 biomolecules-14-00030-t003:** IPA of treated and dry eye versus healthy patients. Numbers and percentages of dysregulated pathways in glaucoma-treated and dry eye syndrome patients compared to those in control groups.

	Treated vs. Control	Dry Eye vs. Control
Host defense and immune system	19 (79%)	13 (39%)
Cellular proliferation and differentiation	3 (13%)	5 (15%)
Mitochondrial dysfunction	-	5 (16%)
Cellular movement and integrity	-	1 (3%)
Wound healing	-	6 (18%)
Unspecific	2 (8%)	3 (9%)
Total	24 (100%)	33 (100%)

**Table 4 biomolecules-14-00030-t004:** Analysis (Z-score) of dysregulated pathways in glaucoma-treated patients versus controls and in dry eye syndrome patients versus controls.

Canonical Pathways	Treated vs. Control	Dry Eye vs. Control
Tec kinase signaling	2.2360	−1.2650
Leukocyte extravasation signaling	1.6330	−1.0000
Integrin signaling	1.3420	−2.3090
TREM1 signaling	1.3420	-
Neuroinflammation signaling pathway	1.1340	−0.4470
Dendritic cell maturation	1.1340	-
Role of NFAT in regulation of the immune response	0.8160	−0.4470
Sirtuin signaling pathway	0.3780	−1.0000
PAK signaling	-	−2.6460
Ephrin receptor signaling	-	−2.5300
Paxillin signaling	-	−2.4490
Regulation of Cellular Mechanics by Calpain Protease	-	−2.4490
TGF-β signaling	-	−2.2360
Agrin interactions at neuromuscular junction	-	−2.2360
Cholecystokinin/gastrin-mediated signaling	-	−1.8900
Actin nucleation by ARP-WASP complex	-	−1.6330
NRF2-mediated oxidative stress response	-	1.5080
G beta gamma signaling	-	−1.1340
ErbB signaling	-	−1.1340
Renin-angiotensin signaling	-	−1.1340
HGF signaling	-	−1.0000
EIF2 signaling	-	1.0000
HIPPO signaling	-	0.8160
IGF-1 signaling	-	−0.8160
PTEN signaling	-	0.7070
ERK/MAPK signaling	-	−0.6320
EGF signaling	-	−0.4470
Sumoylation pathway	-	0.4470
Cdc42 signaling	-	−0.4470
Regulation of eIF4 and p70S6K signaling	-	−0.4470
ErbB4 signaling	-	−0.4470
Neuregulin signaling	-	−0.4470
α-Adrenergic signaling	-	0.4470
Fc Epsilon RI signaling	-	−0.3780

## Data Availability

Data are available upon a reasonable request to the corresponding author.

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
