# Peer review of "Comparison of Transcriptomic Analysis of the Conjunctiva in Glaucoma-Treated Eyes with Dry Eyes and Healthy Controls"

_biomolecules, 2023, doi:10.3390/biom14010030_

Round 1

Reviewer 1 Report

Comments and Suggestions for Authors

In general, this is a comprehensive study with clear rationales and experimental procedures. Most of scientific interpretations of the results and discussion are reasonable. However, the data analysis can be confusing. The high variability of RT-PCR data makes it difficult to interpret and draw the conclusion. Thus, to the reviewer’s point, the results do not fully support the conclusion, except that clearly, different mechanisms are involved in dry eye condition vs glaucoma treatment induced ocular surface disease though. Reorganizing the results and expanding discussion sections with up-to-date references would improve the manuscript quality. Specific comments are listed below.

1.       Most recent references are published in 2020 and not up to date. Did the authors ever compare the current data with Human Eye Transcriptome Atlas database? Few recent review articles also discussed gene expression data around glaucoma treatment (e.g., PMID 35757536) and it would be nice to incorporate some discussion in this manuscript.

2.       Line 60 - although it is obvious, it can be helpful to define dry eye syndrome is considered one of the common OSD.

3.       Several places in this manuscript are missing words. e.g., Line 79 - fibroblast growth factor -??

4.       Consider showing the range and the medium of the age in Table 1, rather than the average with deviation. Is the deviation shown as SD or SEM? 

5.       RT-PCR method is missing in Section 2.2. It would be necessary to show how many samples were used, how to calculate or normalize the expression level (absolute or relative) and statistical analysis.

6.       Consider splitting column "up/down" into two columns in Table 2 to show up and down separately.

7.       Figure 1A: The X-axis title is cut off. Figure 1B: The bottom of volcano lined up at -1. Can you provide the reason?

8.       Section 3.2 - How was the p-value generated? Based on the graphs, most of the differences in gene expression levels seem to be not statistically significant due to the large variability. Thus it is hard to convince the reviewer (the readers) regarding how the data and the description in the main text were validated. Are these p-values based on RT-PCR data or IPA results? Lack of information about RT-PCR procedure and statistical analysis lowered the significance of this work. The statistical issue must be addressed.

9.       Table 3 title is unclear to the reviewer. Please also define the values - are those representing number of genes or pathways or something else?

10.   It looks like Fig 2A was created using Qiagen tool. However, this was not described in the Method section and should be added. The copy right beneath the bar graph should be removed.

11.   Strongly suggest removing Dry eye data from Fig 2B and removing glaucoma data from Fig 3 to avoid confusion. The font size is too small to read, and figures are fuzzy. Please provide new entire figures 2 and 3 with higher resolution. also recommend re-organizing Fig 2B by pathway following the similar format of Fig 3B-E to align with the flow of the text associated with. What is the unit for Y-axis? Please define the asterisks in Fig 2 and 3.

12.   Fig3D, p-value for NCK2 is missing.

13.   What kind of analysis is it in Table 4? It is unclear the data is based on number of differential genes in certain pathway, p-values or IPA Z-score.

14.   Please clearly describe the sample type in discussion. e.g. Line 266 is for dry eye and Line 295 is for glaucoma.

15.   Suggest indicating that these genes were validated in the present study in figure 4 and 5 legend.

16.   Line 337 - in what sample?

17.   Line 348-49 - Are those additional CIC samples extracted from the same donor pool listed in Table 1 or from totally different donors? The difference can serve different purposes: if same donor pool, the additional testing would be just for technical reproducibility; if different donors, the new data can further increase the biological reproducibility.

18.  Line 354 - This should be published already. However, it is new to examine gene expression profile of glaucoma treatment associated OSD. Suggest revising the statement.

Author Response

Thank you for all the suggestions. Please see the attachment.

Reviewer 2 Report

Comments and Suggestions for Authors

This is an interesting article reporting conjunctival transcriptomic changes in glaucoma and dry eye patients; no structural revisions are needed, a minor English revision (and revision of some Tables) is suggested before possible publication.

Comments on the Quality of English Language

A minor English revision is suggested before possible publication (for example "while dry eye patients, alongside the immune response, other mechanisms.."; "The toxicity associated with CAIs is associated mostly.."; "eyes treated with topical glaucoma medications 1"; "Table S4: dry eye pa-tients"; what AA stands for in Table S1 should be reported.. ). Moreover, many words in the columns of different tables (for example Table S4) have been cut short, and should be therefore revised.

Reviewer 3 Report

Comments and Suggestions for Authors

Low quality study -

- There is unclear hypothesis.

- Design has major lacking, to begin with: inclusion and exclusion criteria are missing. Importantly, it appears that the authors did not take into account presence of dry eye in glaucoma patients (overlapping mechanisms of ocular inflammation through dry eye, active agent in glaucoma meds and BAK). They also do not mention if any of the dry eye patients had glaucoma (or was it excluded?)

- Also, treated group is a combination of all patients who are on any glaucoma drop, which vary significantly on their mechanism of action as mentioned in the introduction.

- There is a big load of data without interpretation and very hard to follow. The significance also very unclear.

- Study lacks any mechanistic data; merely showing changes in the expression of genes in the inflammatory or oxidative stress related pathways does not add to the clinical knowledge of presence of inflammation and ongoing oxidative damage in dry eye or other ocular surface inflammatory conditions such as that induced by eye drops. 

- Results are extremely hard to follow and lack proper classification and interpretation. 

- Discussion should be written such that it would interpret results and explain differences between groups and what is found in prior studies. Also need to clearly state what is lacking in this study/limitations. 

Author Response

(The authors gave the same response as above.)

Round 2

Reviewer 1 Report

Comments and Suggestions for Authors

The reviewer greatly appreciated the efforts that the authors had taken in revising the manuscript and responding to the comments. The overall quality of revised manuscript has been improved. Several additional minor comments - 

1. Noted that Table 2 was messed up by additional three values overlapping on the right hand side. Same for Table 4 and suppl. table 1 with few ghost values. Looks like they are line numbers when generating the pdf document.

2. Line 234-235 - second sentence of the table 3 title requires grammar correction.

3. It is very nice presentation in the new Figure 2. However, it appears that new Figure 2 and 3 should be swapped. Figure 3 is missing the same bar legend as shown  in figure 2

4.  In general, it would be not necessary to mention figures or tables in Discussion unless it's a new dataset (e.g., Figure 4 and supplemtary data). If the author do prefer, please correct the figure as Figures 2 and 3 no longer included specific gene expression.

5. Figure S3 and S4 should be swapped with Figure S1 and S2 since they were mentioned earlier in the main text.

6. In many places, Fig 2 and Fig 3 are not applicable anymore due to the changes to suppl Fig. Please correct accordingly throughout the text.

Author Response

Thank you for the suggestions, please see the attachment.

Reviewer 3 Report

Comments and Suggestions for Authors

Thank you for the edits base on comments.

Hypothesis should still be a more clear statement, and should not be "revolving around a possibility" (line 101). It should clearly state what you want to study; in this case from what you have written the hypothesis would be: we hypothesize that the underlying mechanisms of dry eye disease are different from glaucoma treatment related ocular surface disease. The problem with this statement is that dry eye disease is a separate condition from ocular surface toxicity from BAK. A more appropriate way to state the hypothesis could be: we hypothesize that ocular surface inflammation secondary to toxicity from BAK is resulted from molecular and cellular pathways different from other ocular surface inflammatory conditions such as dry eye disease. Then you can say: to test this hypothesis we performed transcriptomic analysis of samples from conjucntivae of patients on glaucoma medications containing BAK and compared that to samples from dry eye patients and healthy controls. 

Title could also be adjusted to: Comparison of transcriptomic analysis of the conjunctiva in glaucoma-treated eyes with dry eyes and healthy controls

Comments on the Quality of English Language

Minor English editing needed including adjusting use of punctuations.

Author Response

(The authors gave the same response as above.)
